# Mechanistic model of hormonal contraception

**A. Armean Wright**[1], **Ghassan N. Fayad**[2], **James F. Selgrade**[1], **Mette S. Olufsen**[1]*

**1** Department of Mathematics and Program in Biomathematics, North Carolina State University, Raleigh, North Carolina, United States of America, **2** Merck & Co., Inc., Kenilworth, New Jersey, United States of America

* msolufse@ncsu.edu

**Data Availability Statement:** This manuscript uses data that are extracted from previously published studies. Figure 1 in Welt CK, McNicholl DJ, Taylor AE, Hall JE. Female reproductive aging is marked by decreased secretion of dimeric inhibin. J Clin

## Abstract

Contraceptive drugs intended for family planning are used by the majority of married or in-union women in almost all regions of the world. The two most prevalent types of hormones associated with contraception are synthetic estrogens and progestins. Hormonal based contraceptives contain a dose of a synthetic progesterone (progestin) or a combination of a progestin and a synthetic estrogen. In this study we use mathematical modeling to understand better how these contraceptive paradigms prevent ovulation, special focus is on understanding how changes in dose impact hormonal cycling. To explain this phenomenon, we added two autocrine mechanisms essential to achieve contraception within our previous menstrual cycle models. This new model predicts mean daily blood concentrations of key hormones during a contraceptive state achieved by administering progestins, synthetic estrogens, or a combined treatment. Model outputs are compared with data from two clinical trials: one for a progestin only treatment and one for a combined hormonal treatment. Results show that contraception can be achieved with synthetic estrogen, with progestin, and by combining the two hormones. An advantage of the combined treatment is that a contraceptive state can be obtained at a lower dose of each hormone. The model studied here is qualitative in nature, but can be coupled with a pharmacokinetic/pharamacodynamic (PKPD) model providing the ability to fit exogenous inputs to specific bioavailability and affinity. A model of this type may allow insight into a specific drug's effects, which has potential to be useful in the pre-clinical trial stage identifying the lowest dose required to achieve contraception.

## Author summary

This study presents a mathematical model for hormonal control of the menstrual cycle of adult women with special emphasis on the effects of oral contraceptive drugs. Our model predicts daily blood levels of ovarian and pituitary hormones in close agreement with data found in the biological literature for normally cycling women. In particular, we study reproductive hormones which are produced by the pituitary gland in the brain and which promote the development of ovarian follicles and the production of ovarian hormones. In

Endocrinol Metab. 1999;84:105-111. Figure 1 and Table 3 in Obruca A, Korver T, Huber, J, Killick SR, Landgren B, Strujis MJ. Ovarian function during and after treatment with the new progestagen Org 30659. Fertil Steril. 2001;76:108-115. Figure 1 and Table 3 in Mulders TMT, Dieben TOM. Use of the novel combined contraceptive vaginal ring NuvaRing for ovulation inhibition. Fertil Steril. 2001;75:865-870.

**Funding:** JFS and AAW were funded in part by the National Science Foundation Division of Mathematical Sciences (DMS) award number 1225607 (https://www.nsf.gov/div/index.jsp?div=DMS). The funders had no role in study design, data collection and analysis, decision to publish, or preparation of the manuscript.

**Competing interests:** I have read the journal's policy and the authors of this manuscript have the following competing interests: GF is an employee of Merck Sharp & Dohme Corp., a subsidiary of Merck & Co., Kenilworth, NJ, USA, and may own stock or stock options in the Company.

turn, the ovarian hormones affect the synthesis and release of the pituitary hormones. We use this model to test the effects of different oral contraceptive treatments. We show that the administration of synthetic progesterone or of synthetic estrogen have a contraceptive effect by preventing ovulation. We illustrate that low doses of each drug given together are most effective at achieving contraception. In addition, model simulations indicate how quickly a combined contraceptive treatment produces a non-ovulatory menstrual cycle and how fast the cycle returns to normal after the treatment ends. If we couple our model with a model for absorption and metabolism of oral contraceptive drugs, the resulting model may help discover minimal effective doses of these drugs and may lead to patient-specific dosing strategies.

## Introduction

The menstrual cycle involves a complex interaction between the ovaries and the hypothalamus and pituitary in the brain. During the cycle, gonadotropin releasing hormone (GnRH) produced by the hypothalamus and ovarian hormones affect the anterior pituitary. In response the pituitary releases gonadotropins including luteinizing hormone (LH) and follicular stimulating hormone (FSH). These gonadotropins stimulate the ovarian system controlling follicle growth and hormone production. The hormones produced by the follicles, notably estradiol (E2), progesterone (P4), and inhibin A (InhA), feedback onto the brain influencing pituitary hormone production [1].

Hormonal contraception has been in development since the early 20th century with the first FDA approved contraception appearing in 1960 [2, 3]. Hormonal contraceptives were mainly composed of synthetic progesterone (a progestin) or a progestin and a synthetic estrogen such as ethinyl estradiol. If these hormones are introduced individually, each can cause contraceptive effects, but high doses of hormonal contraceptives increase the risk for cardiovascular events most notable venous thromboembolism (VTE) and myocardial infarction (MI) [3–10]. Combined hormonal contraceptives (ethinyl estradiol and progestin together) were discovered during testing of a progestin based contraceptive that was accidentally contaminated with a form of estrogen and shown to increase cycle stability and decrease unwanted bleeding patterns [11]. One way to study the effect of administering a combined dose is to use mathematical modeling, which can provide additional insight into effects of varying progestin and synthetic estrogen type and dose.

A number of mathematical models capture dynamics of normal cycling, many of which are based on the formulation by Schlosser and Selgrade [12, 13]. These models are on a time scale of days and predict mean levels of hormone [14, 15]. To our knowledge there have been no adaptations of mathematical models to predict contraceptive effects of exogenous progesterone and estrogen. Specifically, the original menstrual cycle models by Clark et al. [14] and Margolskee and Selgrade [15] do not include ovarian autocrine effects, and therefore they cannot predict the contraceptive response to exogenous administration of progestins.

The model developed in this study, including ovarian autocrine effects, is used to test hormonal contraceptive treatments via oral administration of ethinyl estradiol and progestin. These treatments are modeled by modifying state variables for blood concentrations of E2 and P4. The model does not include a pharmacodynamic component determining how much or how long it takes for specific amounts of oral contraceptives to produce specific changes in the amount of E2 and P4. Therefore, we administer ethinyl estradiol and progestin as

concentrations and assume that contraception is attained if model simulations show a reduction in the LH surge to non-ovulatory levels and/or in P4 levels throughout the cycle.

Model simulations confirm that low and high doses of exogenous progestin reduce the LH surge to non-ovulatory levels as suggested by clinical data [16, 17] for progestin treatments. Also low and high doses of exogenous estrogen reduce the LH surge to non-ovulatory levels. And a combination of low dose estrogen and low dose progestin administered together result in constant non-ovulatory hormone levels. The model may be used to predict which dosing levels of estrogen and progestin produce contraceptive cycles. In addition, model simulations indicate how quickly a combined contraceptive treatment produces a non-ovulatory menstrual cycle and how fast the cycle returns to normal after the treatment ends. These simulations were done to motivate clinical experimentation with similar contraceptive combinations.

## Methods

In this section we discuss the hormonal characteristics of the menstrual cycle, important mechanisms contributing to a contraceptive state, and mechanistic modeling of both the normal menstrual cycle and contraception. Parameter values and dimensions used in the model are found in Table 1.

### The normal menstrual cycle

This section outlines phases and contributing hormones associated with a normal menstrual cycle [1]. Fig 1 depicts the phases of the menstrual cycle and the production and the action of associated hormones. The menstrual cycle arises from a complex interaction between the hypothalamus and the pituitary in the brain, and the ovaries. In the brain a system of capillaries forms a small portal system of blood flow from the hypothalamus to the anterior pituitary, which gives the hypothalamus the means for communicating with the pituitary in absence of a direct neural connection. The hypothalamus secretes GnRH into the portal system stimulated

**Table 1. Parameter values.**

| Hypothalamus/Pituitary | | | Ovaries | | | | | |
|---|---|---|---|---|---|---|---|---|
| **Name** | **Value** | **Unit** | **Name** | **Value** | **Unit** | **Name** | **Value** | **Unit** |
| $V_{0,LH}$ | 500 | IU/day | $\mathbf{b}$ | 0.34 | $\frac{L}{IU}\frac{\mu g}{day}$ | $e_0$ | 30 | pg/mL |
| $V_{1,LH}$ | 4500 | IU/day | $^{*}Ki_{RcF,P}$ | 1 | mL/ng | $\mathbf{e_1}$ | 0.30 | $L^{-1}$ |
| $\mathbf{Km_{LH}}$ | 175 | mL/pg | $^{*}\boldsymbol{\xi}$ | 2.2 | (none) | $\mathbf{e_2}$ | 0.80 | $L^{-1}$ |
| $Ki_{LH,P}$ | 12.2 | mL/ng | $\mathbf{c_1}$ | 0.25 | $\frac{L}{IU}\frac{1}{day}$ | $\mathbf{e_3}$ | 1.67 | $L^{-1}$ |
| $k_{LH}$ | 2.42 | $day^{-1}$ | $c_2$ | 0.07 | $\left(\frac{L}{IU}\right)^{\alpha}\frac{1}{day}$ | $\mathbf{p_0}$ | 0.8 | ng/mL |
| $c_{LH,P}$ | 0.26 | mL/ng | $c_3$ | 0.027 | $\frac{L}{IU}\frac{1}{day}$ | $\mathbf{p_1}$ | 0.15 | $kL^{-1}$ |
| $c_{LH,E}$ | 0.004 | mL/pg | $c_4$ | 0.51 | $\left(\frac{L}{IU}\right)^{\gamma}\frac{1}{day}$ | $\mathbf{p_2}$ | 0.13 | $kL^{-1}$ |
| $a_{LH}$ | 14 | $day^{-1}$ | $d_1$ | 0.5 | $day^{-1}$ | $^{*}Km_{Papp}$ | 75 | mL/ng |
| $v_{FSH}$ | 375 | IU/day | $d_2$ | 0.56 | $day^{-1}$ | $^{*}\mu$ | 8 | (none) |
| $\tau$ | 1.5 | day | $\mathbf{k_1}$ | 0.69 | $day^{-1}$ | $h_0$ | 0.4 | IU/mL |
| $\mathbf{Ki_{FSH,InhA}}$ | 1.75 | IU/mL | $\mathbf{k_2}$ | 0.86 | $day^{-1}$ | $h_1$ | 0.009 | IU/($\mu$gmL) |
| $k_{FSH}$ | 1.9 | $day^{-1}$ | $k_3$ | 0.85 | $day^{-1}$ | $h_2$ | 0.029 | IU/($\mu$gmL) |
| $c_{FSH,P}$ | 12 | mL/ng | $k_4$ | 0.85 | $day^{-1}$ | $h_3$ | 0.018 | IU/($\mu$gmL) |
| $c_{FSH,E}$ | 0.0018 | $mL^2/pg^2$ | $\alpha$ | 0.79 | (none) | $p_{dose}$ | 0, 0.6, or 1.3 | ng/mL |
| $a_{FSH,E}$ | 8.21 | $day^{-1}$ | $\gamma$ | 0.02 | (none) | $e_{dose}$ | 0, 40, or 92 | pg/mL |
| $v$ | 2.5 | L | | | | | | |

Parameters new to this model are marked with a $^{*}$

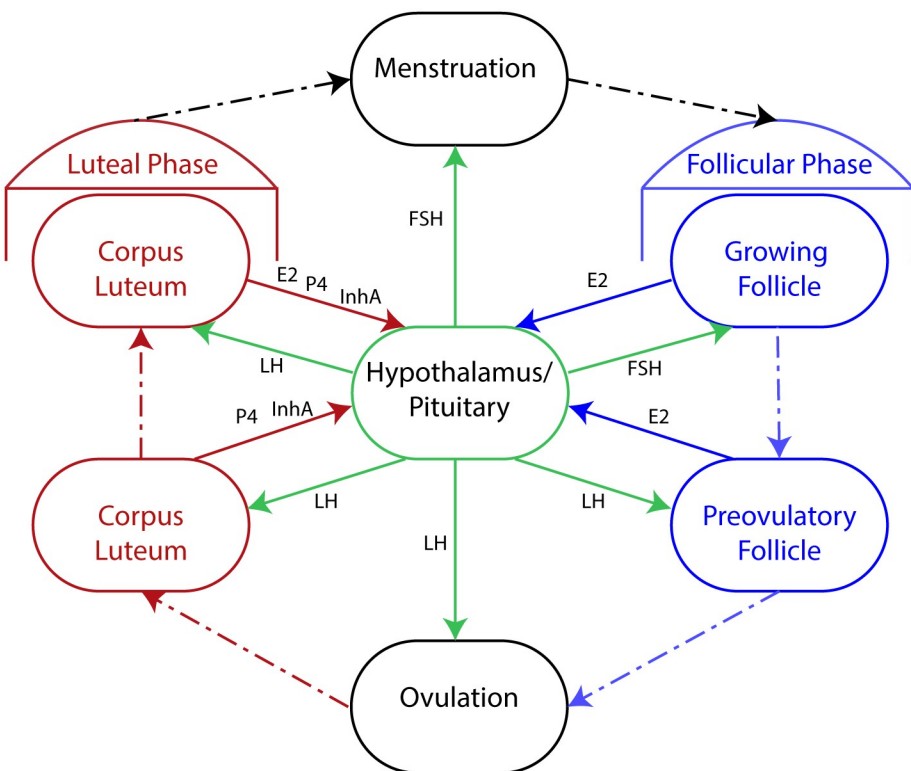

**Fig 1. Menstrual cycle phases.** The outer ring represents ovarian development starting with menstruation (black oval, top) and proceeding in a clockwise direction through the follicular phase (blue ovals, right), ovulation (black oval, bottom), and the luteal phase (red ovals, left). The ovary produces estrodial (E2), progesterone (P4), and inhibin A (InhA), while the brain produces FSH and LH. Directed solid arrows indicate the action of hormones at various stages of the cycle.

by ovarian hormones and feedback from pituitary gonadotropins, and because the half-life of GnRH is short (2-4 minutes), it is only in the portal system that effective levels are found. To facilitate this, GnRH is released in a pulsatile fashion that changes in both magnitude and frequency in response to feedback from ovarian hormones and pituitary gonadotropins. These pulsatile secretions are relatively fast taking place every 1-3 hours depending on menstrual cycle phase, and a wide variability exists amongst individuals [1]. The pulsatile stimulation from GnRH influences the anterior pituitary synthesis and release of gonadotropins (including LH and FSH) influencing the ovaries.

The menstrual cycle consists of two phases (see Fig 1): the follicular phase and the luteal phase. In the weeks before a woman is born her ovaries produce a large mass of germ cells (6-7 million); no more will be produced during her lifetime. Germ cells are transformed by mitosis and a meiotic division into an oocyte. Pre-granulosa cells envelop an oocyte and the resulting unit is called a primordial follicle. This process will eventually happen to all oocytes. As the follicle grows and the surrounding granulosa cell layer proliferates, it becomes a primary follicle. From primary follicle stage it is believed that about 85 days pass before ovulation; most of this time is spent without the influence of pituitary gonadotropins.

The follicular phase begins when multiple follicles are "recruited" and begin expressing FSH receptors, which when stimulated support follicle growth. Further growth leads to LH receptor expression facilitating follicle production of ovarian hormones (including P4, E2, and InhA). The follicles compete for FSH (and later LH) and one or more follicles will advance to

ovulation, if sufficiently stimulated by gonadotropins. Follicles may be arrested at any point during this process through a process of atresia consisting of a break down in granulosa activity eventually ending in apoptosis. All follicles that are "recruited" and unable to reach ovulation will go through this process. During ovulation the follicle ruptures and releases the oocyte through complex mechanisms, for details see [1]. The granulosa cells on the ruptured follicle are luteinized and the structure becomes the corpus luteum. This marks the beginning of the luteal phase during which the oocyte (now called an ovum) is ready for fertilization and the corpus luteum produces P4 and E2 with support from low levels of LH. After about 14 days if fertilization has not taken place, menstruation occurs and the cycle begins again.

E2 has a two stage effect on pituitary LH: at low levels LH release is inhibited but at a certain concentration E2 triggers a massive production of LH. In the beginning of the follicular phase, follicles produce small amounts of E2 inhibiting LH release. As a follicle develops into a dominant follicle, it begins producing E2 in much larger quantities until the second stage effect of E2 causes large amounts of LH to be produced. This mid-cycle rise in LH is called the LH surge and marks the end of the follicular phase. This massive change in synthesis stimulated by E2 is necessary for ovulation. Near the end of the follicular phase InhA is secreted by the follicles, which reduces production of FSH and aids entering the luteal phase. After ovulation the corpus luteum begins producing large amounts of P4 and InhA. In the late luteal phase production of P4 and InhA decreases allowing for increase FSH and E2 production; both priming the cycle for the next follicular phase.

## Contraceptive mechanisms

There is not a specific clinical marker of contraception although many indicators can be used: lack of LH surge, a lack of rise in luteal phase progesterone, or incomplete follicle development. Since it is not feasible to determine cycle timing if ovulation does not occur, contraceptive studies measure progesterone daily, and if P4< 5 ng/mL it is assumed that ovulation is suppressed [16], i.e., P4 can be used as a surrogate marker for contraceptive efficacy. Low progesterone levels or the absence of an LH surge indicates that ovulation and luteinization have not occurred or have not occurred properly. In addition to hormonal effects [16, 18], there are physical indicators such as increased cervical mucus viscosity, which can prevent sperm mobility leading to a contraceptive state.

Physiologically, progestin can cause a contraceptive state through multiple mechanisms. The primary is prevention of ovulation, but secondary effects such as thickening of cervical mucus also cause a contraceptive state [16, 19, 20]. According to [16] ovulation prevention occurs if there is not enough estradiol production to stimulate positive feedback mechanisms necessary for the LH surge. The lack of estradiol production is due to poor follicle development from inadequate LH and FSH support. Progestin reduces LH synthesis directly [1] and by limiting follicular sensitivity to FSH in the early follicular phase [16, 21]. The original models by Clark et al. [14] and Margolskee and Selgrade [15] include progesterone's effect on LH synthesis but do not include progesterone's limiting effect on follicle development via sensitivity to FSH. As a result the original models fail to reproduce contraceptive behavior when administration of exogenous progestins. To capture this effect, our new model introduces a growth limiting factor affecting the early follicle development. Similar to progestins, estrogens act through multiple mechanisms. The primary mechanisms are suppression of LH release by the pituitary [22], included in the models [14, 15], and bolstering progesterone's contraceptive effect [1, 23], which is not accounted for in the original models. In the model presented here, the latter is included by multiplying progesterone, P4, by an increasing function of estrogen, E2.

The major mechanism of progestin in a combined hormonal contraceptive treatment is the same, limiting the sensitivity of follicles to gonadotropins. Estrogen serves two purposes: to limit gonadotropin secretion from the pituitary [1, 24], which is effective enough to cause a contraceptive state from estrogen only dosing, and to increase progesterone receptor expression, which increases progesterone's effectiveness [1].

In summary, contraception can be achieved either by administering exogenous progestins, estrogen, or a combination of the two. This study uses modeling to illustrate that the combined treatment is advantageous because contraception can be achieved by administrating significantly lower doses of each hormone.

## Data

Data used in this study were extracted from previously published studies [16, 18, 25]. Time-series data were digitized and mean values extracted from published Figures and Tables.

Time-series data and error bars for the normal menstrual cycle are digitized from Figure 1 in Welt et al. [25]. These data report mean daily hormone values and variation for FSH, LH, E2, P4, and InhA for 28 days averaged over a group of 23 normally cycling women. These data, repeated over three cycles are shown in Fig 3 together with baseline modeling results.

Data for progestin based contraception are taken from Figure 1 and Table 3 in Obruca al. [16]. These data include hormone values for FSH, LH, E2, and P4 for three doses of Org 30659 progestin administered daily for 21 days. From this study, we extract: mean maximum P4, mean E2, mean maximum FSH, mean maximum LH, and corresponding standard deviations over the 21 day treatment period.

The combined hormonal contraception simulations are compared with data from Figure 1 and Table 3 in Mulders and Dieben [18], which tests effectiveness on ovarian function of a vaginal ring, NuvaRing, containing both a progestin and an estrogen. From this study we compare simulations to data for median maximum hormonal levels for FSH, LH, E2, and P4 for subjects receiving combined hormonal treatment administered vaginally.

## Modeling the normal menstrual cycle

Two components are used to form the menstrual cycle model. The first is a lumped model of the hypothalamus and the pituitary, which predicts synthesis and release of gonadotropins based on circulating concentrations of ovarian hormones (E2, InhA, P4). The second includes the ovaries, accounting for ovarian stages in conjunction with auxiliary equations predicting ovarian hormone production. Fig 2 illustrates the two model components dividing the menstrual cycle into multiple stages representing amount of active tissue in each stage. This distribution of active tissue is used to predict production of ovarian hormones.

**Hypothalamus and pituitary model.** The lumped model (Eqs (1)–(4)) of the hypothalamus and pituitary predicts synthesis and release of FSH and LH as a function of serum concentrations of E2, P4, and InhA. Parameter values and dimensions for Eqs (1)–(4) can be found in Table 1. Dynamics of each pituitary hormone consists of two equations (Eqs (1) and (2) for LH and Eqs (3) and (4) for FSH).

The reserve pool of LH ($RP_{LH}$), Eq (1), tracks the mass of stored gonadotropin LH within the pituitary and is composed of two terms: a positive term representing synthesis and a negative term representing release. The synthesis is promoted by estrogen modeled using a Hill function in $E_2$, and inhibited by progesterone contained in the $P_{app}$ term in the denominator. $P_{app}$, defined in Eq (16), is as a product of $P_4$ and scaled progesterone receptor expression. Hence, $P_{app}$ measures the progesterone signal in the system. The release is promoted by $P_{app}$ and inhibited by $E_2$. The blood hormone concentration $LH$ consists of a positive term,

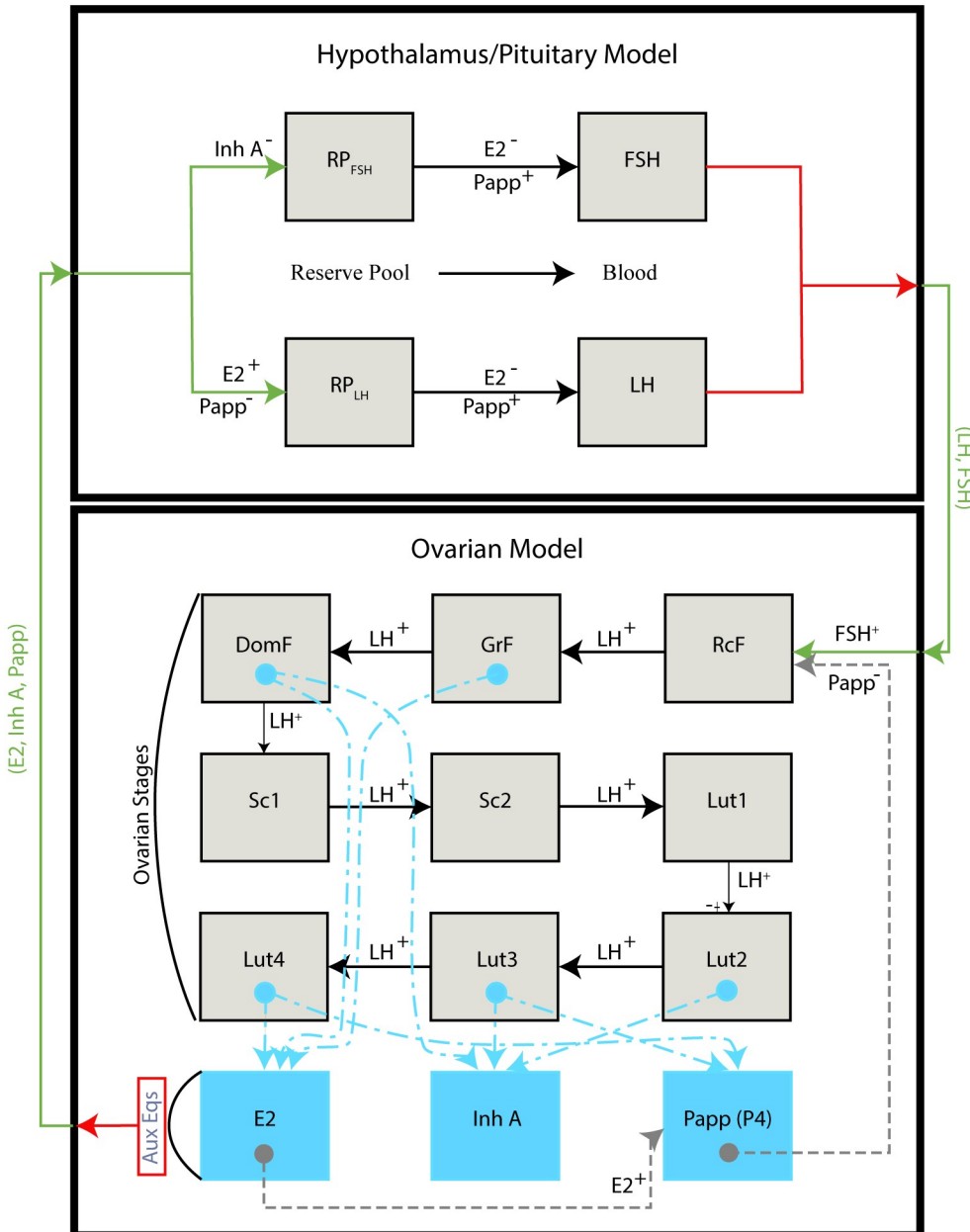

**Fig 2. Full model diagram.** The model diagram shows all states broken into two sub-models. The hypothalamus/pituitary model has four states (RP$_{FSH}$,FSH,RP$_{LH}$, LH) and the ovarian model has nine states (RcF, GrF, DomF, Sc1, Sc2, Lut1-4). In the hypothalamus/pituitary model, the black horizontal arrows represent hormone (E2, Papp) movement and in the ovarian model they represent movement of cells or tissue (mediated by LH) between stages. The red horizontal arrows represent output from a sub-model, and the green horizontal arrows represent input into a sub-model. A hormone $H$ written as $H^+$ or $H^-$ has a stimulating or inhibiting effect respectively on movement between chambers or effectiveness of a hormone within the chamber. The blue dashed-dotted lines within the ovarian model show stages contributing to ovarian hormone production in the auxiliary equations. The gray dashed lines in the ovarian model represent autocrine influence of ovarian hormones within the model. Full expression for the hypothalamus/pituitary and ovarian models can be found in Eqs (1)–(4) and Eqs (5)–(13), respectively, and parameter values and dimensions are listed in Table 1.

denoting the release from the reserve pool scaled by blood volume ($v$), and a negative linear term representing clearance of the hormone from the blood.

Similarly, the reserve pool of FSH ($RP_{FSH}$), Eq 3, tracks the mass of stored gonadotropin FSH within the pituitary. It is also composed of two terms: a synthesis term inhibited by inhibin A ($InhA$) and a release term that similar to $RP_{LH}$ is promoted by $P_{app}$ and inhibited by $E_2$. It should be noted that the biological process for inhibin A's effect is lengthy (it takes approximately 1.5 days for this to have an effect [26]). To include this effect, we introduce a discrete time-delay $InhA(t - \tau)$ in the synthesis term. Finally, the release from the reserve pool $RP_{FSH}$ increases blood $FSH$ levels, and $FSH$ decreases linearly modeled by a clearance term.

$$\frac{d}{dt}RP_{LH} = \frac{V_{0,LH} + \frac{V_{1,LH}E_2^8}{Km_{LH}^8 + E_2^8}}{1 + P_{app}/Ki_{LH,P}} - \frac{k_{LH}[1 + c_{LH,P}P_{app}]RP_{LH}}{1 + c_{LH,E}E_2} \tag{1}$$

$$\frac{d}{dt}LH = \frac{1}{v}\frac{k_{LH}[1 + c_{LH,P}P_{app}]RP_{LH}}{1 + c_{LH,E}E_2} - a_{LH}LH \tag{2}$$

$$\frac{d}{dt}RP_{FSH} = \frac{V_{FSH}}{1 + InhA(t - \tau)/Ki_{FSH,InhA}} - \frac{k_{FSH}[1 + c_{FSH,P}P_{app}]RP_{FSH}}{1 + c_{FSH,E}E_2^2} \tag{3}$$

$$\frac{d}{dt}FSH = \frac{1}{v}\frac{k_{FSH}[1 + c_{FSH,P}P_{app}]RP_{FSH}}{1 + c_{FSH,E}E_2^2} - a_{FSH}FSH \tag{4}$$

During a normal cycle estrogen exhibits a 2-stage effect on LH synthesis. Low levels of estrogen inhibit LH release, high levels strongly stimulate production [22]. This 2-stage behavior is represented by Eq (1): the synthesis term contains a Hill function dependent on E2, which at a critical level of E2 increases LH synthesis, and in the second term E2 inhibits LH release. The Hill function represents the main biological mechanism of the hypothalamus/pituitary model as it is responsible for the mid-cycle LH surge in response to rising E2 levels. As E2 increases above a threshold level, the Hill function in the first term in Eq (1) becomes large enough to produce the priming affect of E2 on LH synthesis.

In Eqs (1)–(4) there are important relationships between E2 and P4 in the synthesis and release of gonadotropins. Although E2 is responsible for simulating synthesis through the Hill function, it also inhibits the release of both LH and FSH which is a secondary mechanism of estrogen based contraception. In the positive term in Eq (1), the denominator contains the secondary contraceptive effect of progestin inhibiting LH synthesis.

**Ovarian model.** The ovarian model tracks sensitive follicle mass as it moves through the biological phases of the menstrual cycle: follicular phase and luteal phase (see Fig 1). To simulate the timing of follicle development, the model breaks each of the phases into multiple stages and adds two compartments as a transition from the follicular to luteal phase labeled $Sc1$ and $Sc2$, which represent two stages of the follicle during ovulation. Ovarian hormone production is derived from the mass in each stage in the auxiliary equations. It is assumed that serum concentrations of the ovarian hormones are at a quasi-steady state, i.e., that the hormone concentration is proportional to the masses. This is an assumption formulated in [27] and used in model construction in the original menstrual cycle model [12, 14], later modified by Margolskee and Selgrade [15].

The full system of equations describing the ovarian stages (Eqs (5)–(13)) and associated auxiliary equations (Eqs (14)–(17)) are adopted from [14, 15]. The follicular phase is broken into 3 stages: recruited follicle ($RcF$), growing follicle ($GrF$), and dominant follicle ($DomF$).

The mass tracked cannot be thought of directly as mass of the follicles, but as mass of follicle contributing to hormone production. Estrogen is produced by follicles in the mid to late follicular phase and in the late luteal phase, so the auxiliary equation for estrogen (Eq (14)) consists of terms proportional to the masses in the $GrF$, $DomF$, and $Lut_4$. Ovulation is broken into two stages and the luteal phase into four stages. Equations calculating ovarian hormones, assumed proportional to masses in different stages, are in Eqs (14)–(17). In Eqs (14) and (15) exogenous doses of estrogen ($e_{dose}$) and progestin ($p_{dose}$) are added, respectively. We assume that the added hormone (progestin or estrogen) acts as the endogenous hormone. Eq (16) defines the variable $P_{app}$, which groups the effects of progesterone on both the brain and the ovaries, i.e., we assume that P4 affects all tissues in the same way. In the ovaries, during follicle development, P4 limits sensitivity to FSH [20, 21] as described by the denominator of Eq (5).

The second term in Eq (16) represents the increase in P4 receptor expression due to E2 [23, 28] in the form of an increasing Hill function. Hence, the presence of estrogen enhances the effectiveness of P4.

The variable $P_{app}$ is used everywhere progesterone has an effect, as discussed in detail in the modeling contraception section below.

$$\frac{d}{dt} RcF = (b + c_1 RcF) \frac{FSH}{(1 + P_{app}/Ki_{RcF,P})^{\xi}} - c_2 LH^{\alpha} RcF \tag{5}$$

$$\frac{d}{dt} GrF = c_2 LH^{\alpha} RcF - c_3 LH \; GrF \tag{6}$$

$$\frac{d}{dt} DomF = c_3 LHGrF - c_4 LH^{\gamma} DomF \tag{7}$$

$$\frac{d}{dt} Sc_1 = c_4 LH^{\gamma} DomF - d_1 Sc_1 \tag{8}$$

$$\frac{d}{dt} Sc_2 = d_1 Sc_1 - d_2 Sc_2 \tag{9}$$

$$\frac{d}{dt} Lut_1 = d_2 Sc_2 - k_1 Lut_1 \tag{10}$$

$$\frac{d}{dt} Lut_2 = k_1 Lut_1 - k_2 Lut_2 \tag{11}$$

$$\frac{d}{dt} Lut_3 = k_2 Lut_2 - k_3 Lut_3 \tag{12}$$

$$\frac{d}{dt} Lut_4 = k_3 Lut_3 - k_4 Lut_4 \tag{13}$$

$$E_2 = e_0 + e_1 GrF + e_2 DomF + e_3 Lut_4 + e_{dose} \tag{14}$$

$$P_4 = p_0 + p_1 Lut_3 + p_2 Lut_4 + p_{dose} \tag{15}$$

$$P_{app} = \frac{P_4}{2}\left(1 + \frac{E_2^\mu}{Km_{P_{app}}^\mu + E_2^\mu}\right) \tag{16}$$

$$InhA = h_0 + h_1 DomF + h_2 Lut_2 + h_3 Lut_3 \tag{17}$$

## Modeling contraception

Progestin and estrogen act through different pathways and mechanisms to cause a contraceptive state. Progestin acts by limiting follicular sensitivity to FSH and by inhibiting LH synthesis, whereas estrogen inhibits LH release. Two important autocrine effects in the model capture the basic dynamics of both combined hormonal contraceptives and progestin only treatments. The first is contained in Eq (5) via inhibition of FSH at the ovarian level due to $P_{app}$. The second, described by Eq (16), enhances the effect of P4 in the presence of estrogen.

It should be noted that the model tracks blood concentrations of ovarian hormones, exogenous progestin and estrogen levels.

Therefore contraceptive "doses" always refer to concentrations. To analyze model dynamics for each contraceptive treatment, model simulations must have reached stable behavior (cyclic or steady state) ensuring that effects of initial conditions have dissipated. To achieve this, we administered the contraceptive drugs three months prior to analyzing simulation results.

The new model studied here is based on the model in Margolskee and Selgrade [15], which cannot predict contraceptive behavior. For instance, if a progestin dose of 1.3 ng/mL is administered to the Margolskee and Selgrade model [15], it results in a slightly higher LH surge than the normal. This occurs because a small additional amount of P4 is more effective at increasing FSH production (see Eq (4)) than inhibiting LH production (see Eq (1)). More FSH causes increased early follicular growth (Eq (5) without the $P_{app}$ term) resulting in more early follicular E2 and hence a slightly higher LH surge. Including the $P_{app}$ term without the E2 enhancement of Eq (16) in Eq (1) dampens this growth and decreases the LH surge but it still is at an ovulatory level. Thus in order to model progestin's contraceptive effect both the inhibition of P4 on early follicular development and the enhancement due to E2 are needed to predict contraception.

In the following we describe how the new model components achieve contraception by progestin, estrogen, and the combined treatments.

**Progestin based contraception.** The major mechanism of interest is the inhibiting effect progestin has on FSH's ability to produce follicular tissue that is sensitive to LH. A secondary effect present in Eq (1) is P4's inhibition of LH synthesis. It is believed that the hormonal contraceptive effect of progestin is inhibiting growth of active follicular tissue during the early follicular phase by reducing follicular sensitivity to FSH [16].

This effect is included by introducing $P_{app}$ in Eq (5), which inhibits *RcF* growth due to FSH. Under normal conditions, the P4 concentration is very low during this part of the cycle, so the inhibitory effect on follicle growth is negligible. Two growth terms proportional to FSH in Eq (5) are divided by a term including the applied progestin. As a result, FSH has less of a stimulatory effect on follicle growth if the applied progestin is high, such as during treatment with

a contraceptive drug or during the luteal phase. These abnormal conditions diminish follicle tissue sensitivity to LH, which inhibits the follicle tissue movement through the normal stages. In addition, the progestin effect is increased by E2 via the Hill function in $P_{app}$ (see Eq (16)). Thus little appreciable follicular mass can reach the growing follicle stage $GrF$, preventing the mid-cycle rise in E2. Without the rise in E2, the LH surge does not happen and ovulation cannot occur.

**Estrogen based contraception.** Estrogen is contraceptive as well. This comes from inhibiting the release of both LH and FSH from the pituitary, modeled by Eqs (1) and (3). The end result is the same as with progestin. Insufficient gonadotropins from the pituitary prevent a LH surge. The addition of estrogen to the treatment allows for a smaller dose of progestin.

**Combined hormonal contraception.** In the combined treatment with estrogen and progestin, estrogen serves to bolster progesterone's effect but also inhibits LH release [1]. The presence of estrogen upregulates progesterone receptor expression, which increases P4's effectiveness. This has been shown in ovine and rat uterine cells [1, 28]. A possible secondary effect of the the mid-cycle rise in estrogen is to prime P4 receptors for the luteal phase [28]. To represent this dynamic we have added Eq (16), which scales circulating P4 in the body with a steep Hill function dependent on estrogen. The resulting $P_{app}$ is used as the active progesterone in the system. Without estrogen $P_{app}$ is half of the produced P4. At a certain level of estrogen the receptor expression is assumed higher and $P_{app}$ approaches P4. In addition, the Hill function in Eq (16) depends on estrogen causing progestin to be effective at lower doses if there is also an estrogen component.

## Model summary

The model described above is formulated as a system of 13 delay differential equations of the form

$$\frac{dx}{dt} = f(x, y, t, t - \tau),$$

where

$$x = \{RP_{LH}, LH, RP_{FSH}(t - \tau), FSH, RcF, GrF, DomF, Sc_1, Sc_2, Lut_1, Lut_2, Lut_3, Lut_4\}$$

with four auxiliary equations for the ovarian hormones

$$y = \{E_2, P_4, P_{app}, InhA\}$$

and 46 parameters given in Table 1. Estimated parameters are marked in bold and the remaining parameters are from [15]. New parameters are marked by a *.

The clearance rate for FSH is from [29]. The clearance rate for LH is from [30]. Equations are solved with MATLAB using the delay differential equation solver (dde23) and bifurcation analysis is done using DDE-BIFTOOL [31].

To analyze this model we conducted four simulations studying:

- The model prediction of baseline hormones without contraception. These results were compared to data repeated over three cycles of length 28 days. Results are compared to data from Figure 1 in Welt et al. [25].

- The response with contraception for low and and high dose of progestin, comparing the response to the normally cycling data from Figure 1 in Welt et al. [25] and the contraceptive state extracted from Figure 1 and Table 3 in [16].

- The response with contraception for low and and high dose of estrogen, comparing the response to the normally cycling data from Figure 1 in Welt et al. [25]. This simulated response is not compared to data.

- The response to combined treatment with low doses of progestin and estrogen. The low doses used here are the same studied in the isolated treatment studies. Results for this simulation is compared to data from Figure 1 and Table 3 in Mulders and Dieben [18].

All simulated results are displayed after the solution has reached stable constant or oscillating behavior.

In addition to forward simulations comparing the response to different dosing strategies, we conduct a bifurcation analysis to determine when the model goes from stable oscillations to steady state with the combined estrogen/progestin contraception. Finally, we study the effect of removing contraception to understand how long it takes to return to normal cycling.

## Results

This study presents a mathematical model of the menstrual cycle that can predict normal cycling as well as the dynamic response to exogenous progestin and estrogen dosing as described in the model summary section.

Fig 3 shows the model's fit to data for normal cycling women digitized from Figure 1 in Welt et al. [25]. The data is for a single cycle and we have concatenated it for the number of cycles necessary to compare simulations. In this study, we use the term "total contraception" to describe a contraceptive treatment which results in model simulation reaching steady state, i.e., all variables become constant. While biological contraception is achieved before total contraception, quantitatively it is useful to look at where total contraception takes place for comparative analysis. In all plots, data are represented by magenta dot-dashed lines and model simulations by solid blue lines. Unless otherwise stated, asymptotic solutions of a stable cycle or a steady state are displayed. Dosing is applied 3 months before time zero and continues throughout the simulations.

Notice that the hormone profiles in Fig 3 are not as close to the Welt data [25] as the profiles in the original model by Clark et al. [14] are to the data in McLachlan et al. [32], which Clark et al. [14] used to identify parameters. This occurs because the Welt data is noisier than the McLachlan data, so parameter identification in Clark et al. [14] is more accurate. However, the McLachlan data does not contain inhibin B which we will use in the future to improve this study.

Model parameters for this study were largely kept at the values used in the study by Margolskee and Selgrade [15], except for changes needed for the new model components. New parameters and parameters associated with contraception were estimated and these are marked with bold in Table 1. The estimated cycle length is 28.65 days, therefore Fig 3 displaying the contraception free result is not completely in phase with the data. We have depicted results aligning the data to the middle of the three cycles. It is possible to adjust the cycle length by scaling the ovarian model parameters $e_i$ and $p_i$, but we chose not to do so to keep the model as close as possible to the original model, and the estimated cycle is still within normal values. Moreover, the objective of this study is to predict the effect of contraception, and the qualitative results discussed below are not dependent on matching the cycle length exactly to the data.

### Progestin based contraception

With the addition of exogenous progestin, the model approaches a contraceptive state in a dose-dependent manner. Data for a contraceptive state due to progestin are taken from

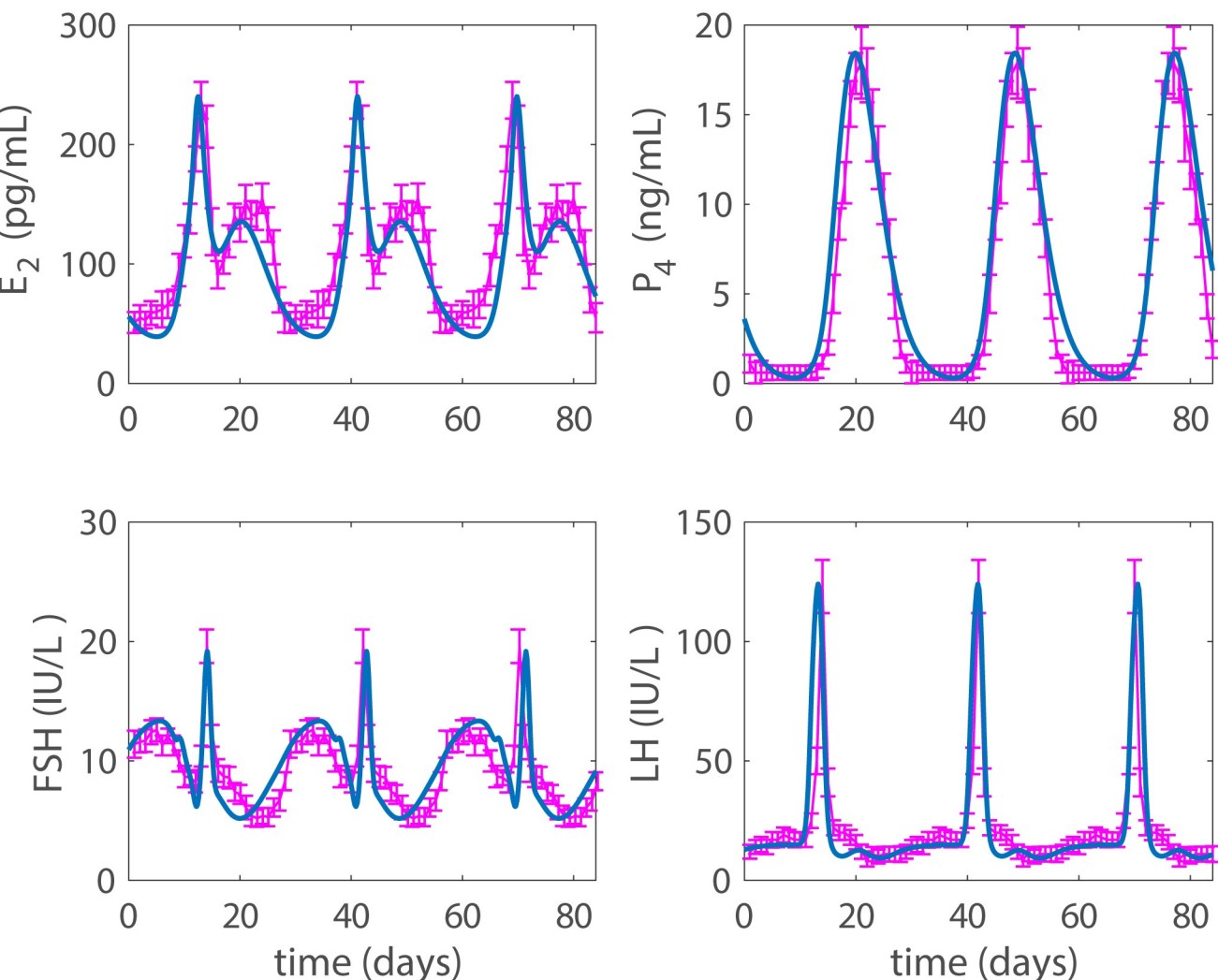

**Fig 3. Normal cycle.** Results for a normal cycle with no exogenous estrogen or progestin. The model output for 84 days (3 cycles) is denoted by the solid blue line and the connected points are data with error bars digitized from Figure 1 in Welt et al. [25].

Figure 1 and Table 3 in Obruca et al. [16]. The data display the mean maximum and standard deviation of the hormonal values after a 21 day treatment of a progestin based contraceptive. The mean maximum value is denoted with the red solid horizontal line and the standard deviation is represented by the red dotted horizontal line in Figs 4 and 5. Data (dashed-dotted magenta lines) from Figure 1 in Welt et al. [25] for a normal cycle are plotted for reference in the figures. Results from both a low and a high dose of progestin are shown (Recall, the model tracks blood concentrations, which we refer to as doses).

Notice that in Figs 4–7 the FSH profiles in response to contraceptive treatments are higher than biologically observed [18]. This occurs in our model because FSH synthesis is suppressed only by inhibin A (see Eq (3)). In our model of contraception, ovulation does not occur so the corpus luteum does not develop and InhA is produced at low levels (see Eq (17)). Hence, a considerable amount of FSH is synthesized and the FSH profile is high. Inhibin B is produced during the follicular phase of the cycle and would provide inhibition of FSH in a contraceptive situation. However, including inhibin B would complicate the present model significantly.

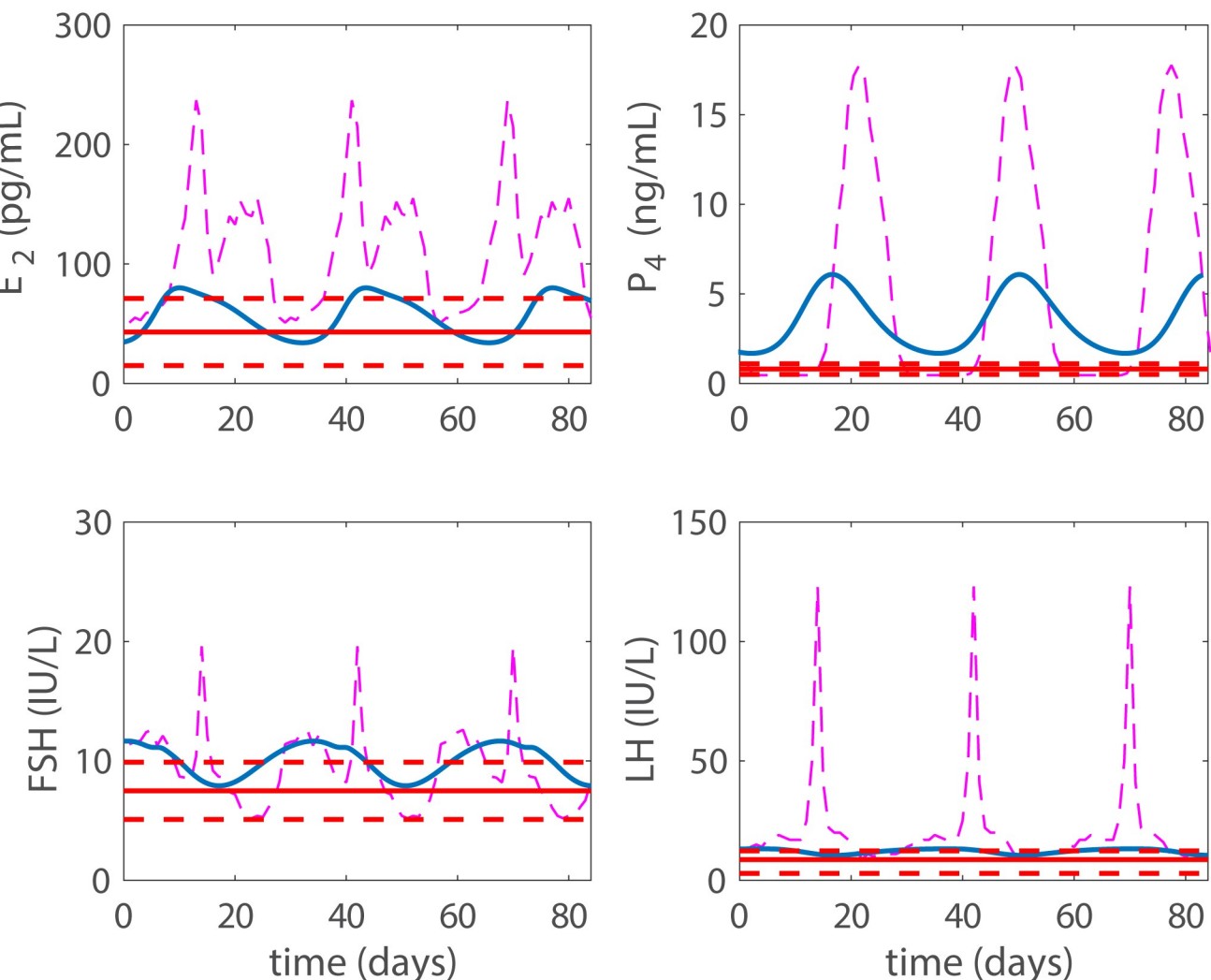

**Fig 4. Progestin low dose.** Model results with a low dose ($p_{dose}$ = 0.6 ng/mL) are plotted with a solid blue line, while the solid red horizontal red line denotes the mean maximum hormonal values resulting from the 21 day progestin treatment reported in Figure 1 and Table 3 in Obruca et al. [16] with the standard deviation represented by the horizontal dashed lines. The mid-cycle LH surge has been eliminated. With this dose we have reached biological contraception by preventing the LH surge, but we have not reached total contraception. For comparison, the normal cycling data are presented by a dashed-dotted magenta line.

The doses giving the hormone levels discussed above from Figure 1 and Table 3 in Obruca et al. [16] are in mg, whereas in the model they are given in concentrations. Approximate concentration doses were obtained by converting the high dose values reported in Table 3, [16]. These were subsequently adjusted to obtain a high dose, representing the lowest concentration giving a constant long-term solution. The low does was set to approximately half the high dose. More specifically, for the high dose $p_{dose}$ = 1.3 ng/mL and for the low dose $p_{dose}$ = 0.6 ng/mL.

This low dose does not result in total contraception, but the LH surge has been effectively eliminated (see Fig 4) likely causing biological contraception. In Fig 5 the high dose case ($p_{dose}$ = 1.3 ng/mL) is displayed and steady state has been reached, i.e., our defined total contraception.

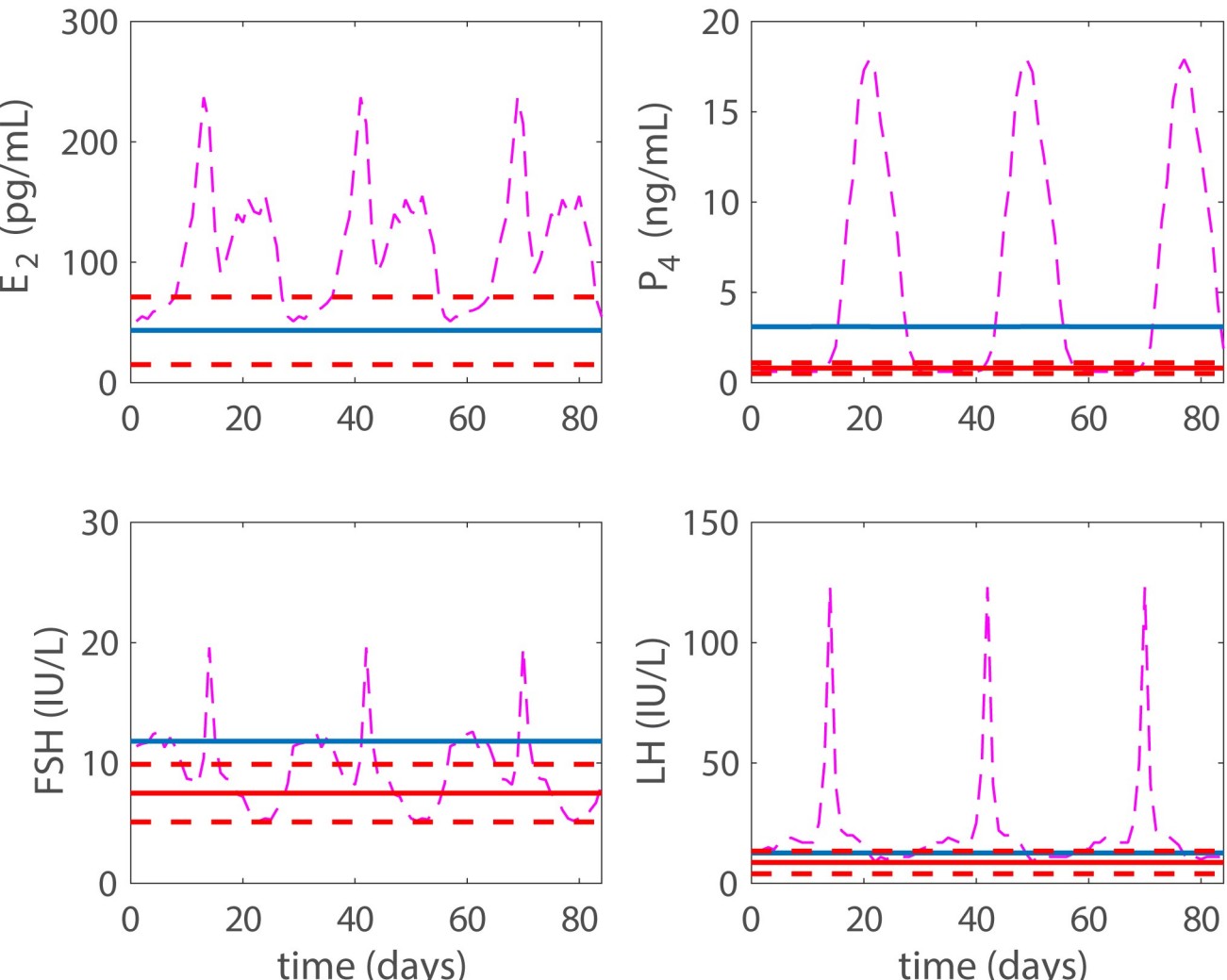

**Fig 5. Progestin high dose.** Model results with high a dose $^*p_{dose}$ = 1.3 ng/mL are plotted with a solid blue line, while the solid red horizontal line represents the mean maximum hormonal value resulting from the 21 day progestin treatment reported in Figure 1 and Table 3 in Obruca et al. [16] with the standard deviation represented by the horizontal dashed lines. For P4 we note a significant difference between model predictions and the data. This likely stems from the fact that in the model P4 includes both endogenous and exogenous progestin, while the data only measure the endogenous levels. We have reached a steady state here and thus total contraception. For comparison, the normal cycling data are presented by a dashed-dotted magenta line.

## Estrogen based contraception

While estrogen only contraceptives are not normally used in practice, a high enough dose of estrogen results in contraception. As with progestin, two cases are considered: a low dose that does not cause total contraception and a higher one that does. The low dose case (40 pg/mL) is depicted in Fig 6. Again, the low dose does not achieve total contraception, but the LH surge has been reduced to a level that likely indicates biological contraception. The dose (92 pg/mL) that accomplishes total contraception is shown in Fig 7. In both figures, we have plotted data from [25] for reference to a normal cycle. Data for estrogen only contraception in humans is unavailable, but hormonal values fall within a reasonable biological range for a contraceptive state.

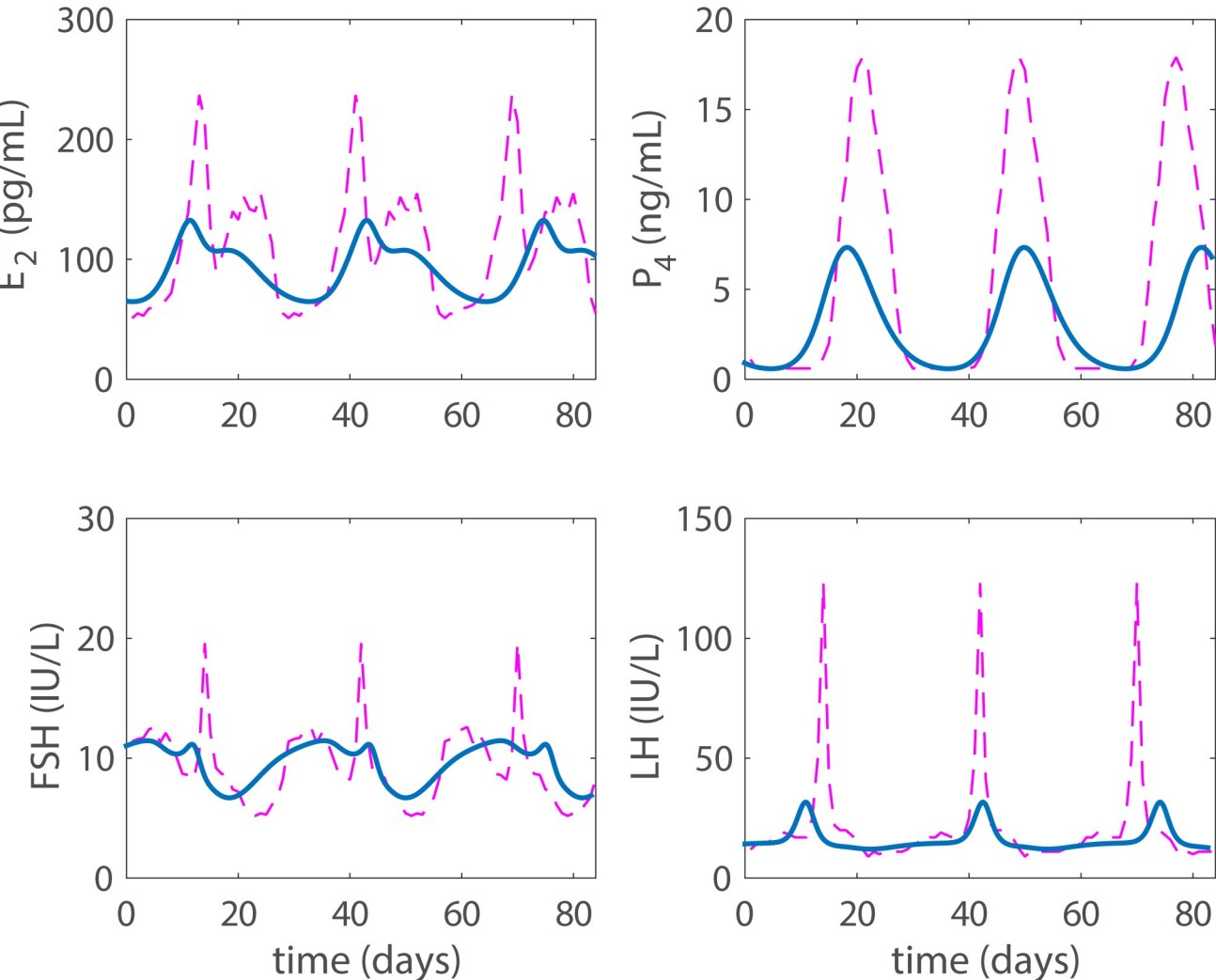

**Fig 6. Estrogen low dose.** Model results (solid blue line) with $e_{dose}$ = 40 pg/mL. In this simulation, LH has a small mid-cycle rise, but the large LH surge is significantly suppressed and ovulation does not occur indicating a contraceptive state, yet the hormone levels still vary during the cycle. For comparison, the normal cycling data are presented by a dashed-dotted magenta line.

## Combined hormonal contraception

Applying the two low doses to the model at the same time yields the results shown in Fig 8. Model hormone predictions are compared with values taken from Figure 1 and Table 3 in Mulders and Dieben [18]. The dotted red horizontal line is the median of the maximum concentration of the hormone between days 8 and 13 of treatment. The solid horizontal line is the predicted hormone concentration output from the model, and for comparison, the normal cycling data are presented by a dashed-dotted magenta line.

## Bifurcation analysis

A bifurcation is a change in qualitative behavior of a system and occurs as a parameter of the system crosses a critical value. A Hopf bifurcation occurs when moving over this critical value causes a change from cyclic behavior to steady state behavior or vice versa. If the model is in a cyclic state, a significant enough increase in $p_{dose}$, $e_{dose}$, or both will move the model over a

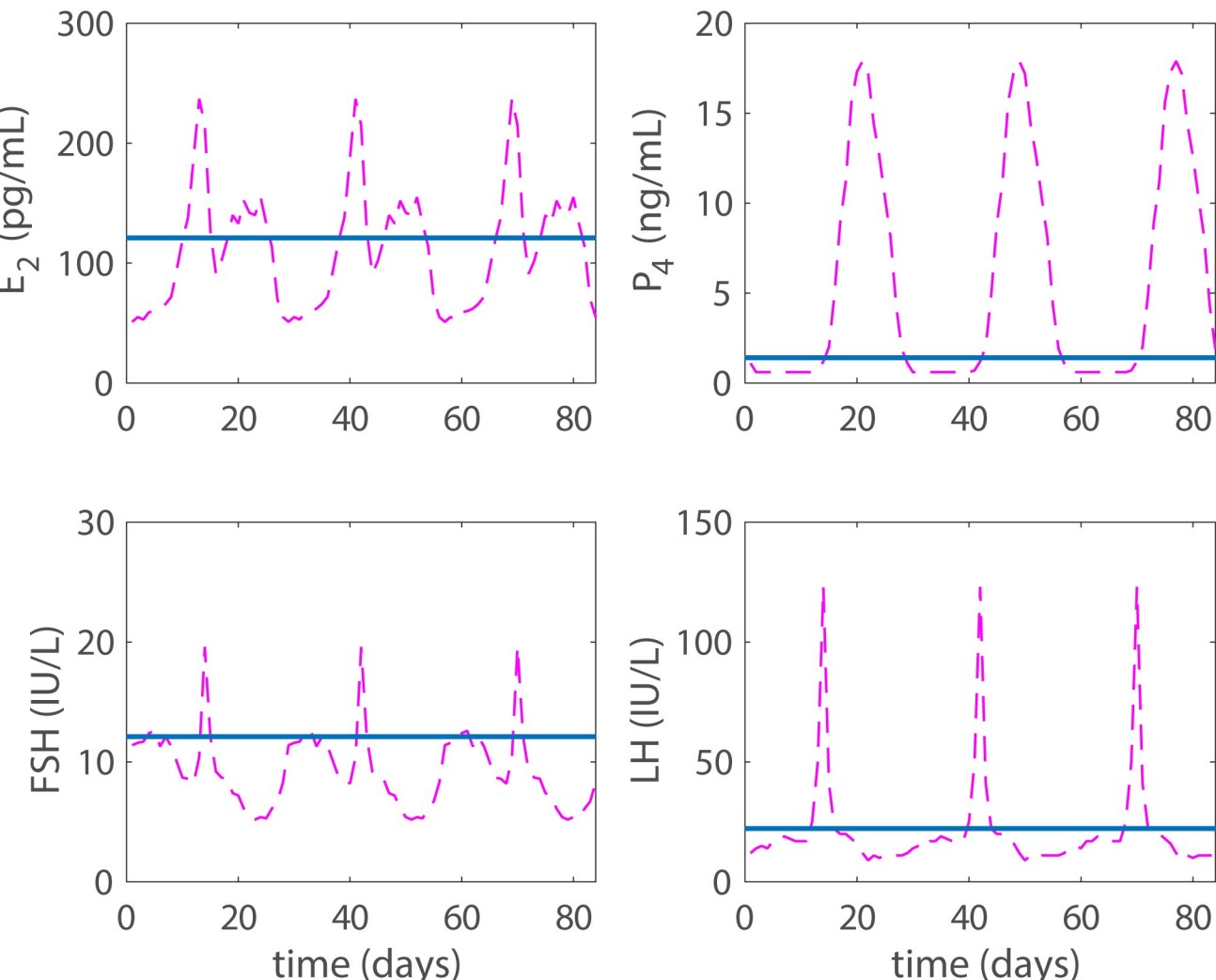

**Fig 7. Estrogen high dose.** Model results (solid blue line) with $e_{dose}$ = 92 pg/mL. For this high dose total contraception has been achieved. For comparison, the normal cycling data are presented by a dashed-dotted magenta line.

Hopf bifurcation from the cyclic region into steady state region. The curve in the ($e_{dose}$, $p_{dose}$) space of Hopf bifurcations then illustrates where total contraception is achieved.

The curve in Fig 9 displays Hopf bifurcations in the ($e_{dose}$, $p_{dose}$) space illustrating the relationship between doses and total contraception. This curve is constructed using the software DDE-BIFTOOL [31], which identifies bifurcations for delay differential equations. We know that if $p_{dose}$ = 0 then the system attains a steady state at $e_{dose}$ = 92 pg/mL, see Fig 7. If $e_{dose}$ is decreased from 92 pg/mL, DDE-BIFTOOL finds the Hopf bifurcation at $e_{dose} \approx$ 90 pg/mL. We fix the $p_{dose}$ parameter at small increments between 0 and 1.3 pg/mL and search for Hopf bifurcation with respect to the parameter $e_{dose}$ to generate the curve of Hopf bifurcations in Fig 9. Below the curve are periodic solutions of the model (cyclic behavior) and above the curve are steady state solutions. The normal state of the model is at (0, 0) where there is no dose of either type. The Hopf bifurcations define the doses at which total contraception takes place: the exact point at which the periodic solution becomes a steady state solution. The high dose cases for estrogen and progestin are shown as stars on the $x$ and $y$ axis respectively. The combination

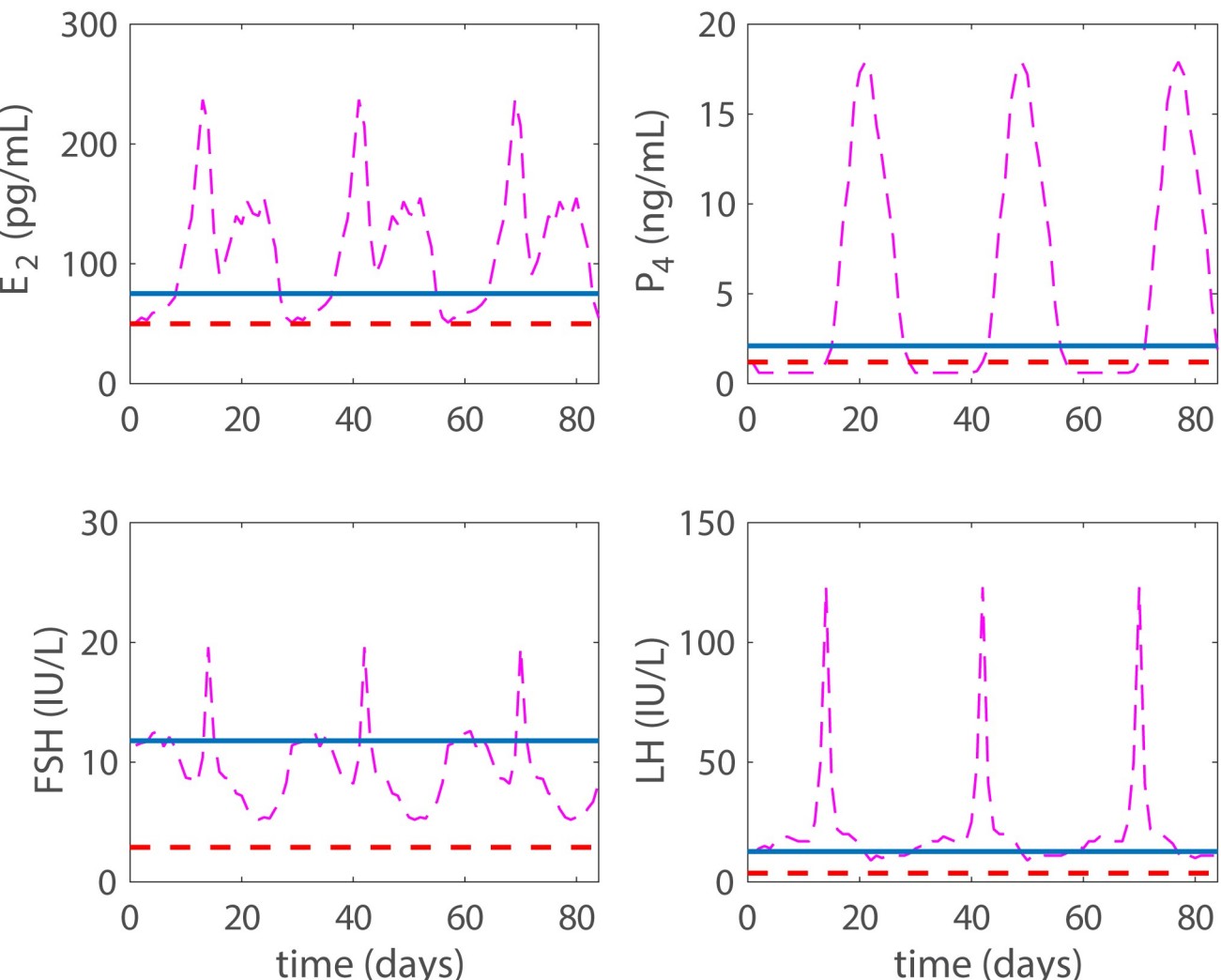

**Fig 8. Combined low dose.** Model results (solid blue line) with $p_{dose}$ = 0.6ng/mL and $e_{dose}$ = 40 pg/mL. The dotted red line is the median maximum hormonal value during days 8-14 of combined hormonal treatment reported in Figure 1 and Table 3 in Mulders and Dieben [18]. These are the two low doses that did not reach total contraception when used individually. The application of both low doses though has achieved total contraception. For comparison, the normal cycling data are presented by a dashed-dotted magenta line.

low dose is marked just above the Hopf curve in the steady state solution space in red. The two high dose treatments can be found along either axis where the Hopf curve intersects: for progestin only at $p_{dose} \approx 1.3$ ng/mL and for estrogen only at $e_{dose} \approx 92$ pg/mL.

## Return to normal cycling

All results presented up to this point have been asymptotic solutions that have allowed time for the model to reach a stable cycle or steady state solution. It is imperative, however, in contraceptive design that introduction of a contraceptive quickly cause a non-ovulatory state and removal of the contraceptive results in return to normal cycling. To demonstrate this behavior the model simulates nine cycles assuming cycles are 28 days. The first three cycles are normal, the next three cycles have a combined low dose of estrogen and progestin, and the last three cycles have the dose in the blood exponentially decaying due to the drug's half-life. Both elimination half-lives of the drugs are short compared to the model time scale: progestin has an

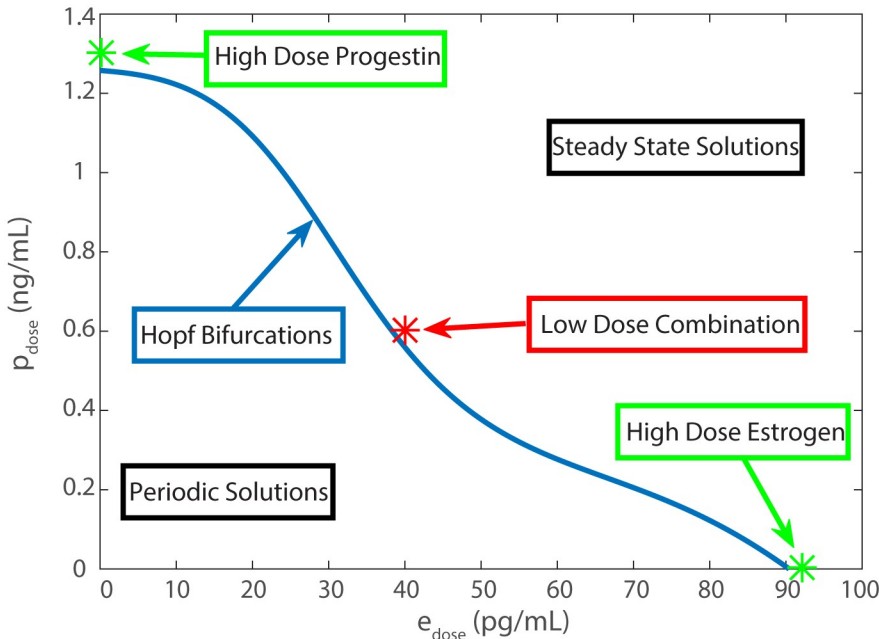

**Fig 9. Hopf bifurcations.** Bifurcation diagram representing location of Hopf bifurcations in the ($e_{dose}$, $p_{dose}$) space. Solutions below the curve of Hopf bifurcations are periodic and solutions above the curve are steady state. Our total contraception as we have defined it then occurs along this curve of Hopf bifurcations. Any doses falling above the curve are totally contraceptive and any below are not. The low dose combination that we tested (used in Fig 8) is shown with a red star and falls just into the steady state region. The progestin (used in Fig 5) and estrogen (used in Fig 7) only doses can be seen approximately where the Hopf curve intersects the axes.

approximate half-life of a day [33, 34] and estrogen of two days [35]. The resulting simulation is shown in Fig 10. The vertical dotted lines represent the beginning and end of dosing. The simulation transitions from a normal cycling state to a contraceptive state and back to normal cycling within one to two cycles of the treatment's removal. The contraceptive portion of the simulation does not have time to reach a steady state, but is completely devoid of an LH surge. The combined dose given is strong enough to cause total contraception if treatment was applied for a longer window.

## Discussion

In this study, we developed a model for menstrual cycle dynamics that can predict the effects of several contraceptive hormone treatments preventing ovulation. New key features in this model are the ovarian autocrine effects: progesterone inhibiting growth of the recruited follicle and estrogen amplifying the effects of progesterone shown in Eqs (5) and (16), respectively. Data from the biological literature [25] are used to identify model parameters and the resulting model simulations approximate well the hormonal profiles of normally cycling women (Fig 3).

Then the model is used to test the effects of five different hormonal contraceptive treatments. It is assumed that the doses of exogenous hormones are added directly to the blood and act as the natural analogues in the body. Low and high doses (concentrations) of exogenous progestin reduce the LH surge to non-ovulatory levels (Figs 4 and 5) and reflect clinical data reported in Figure 1 and Table 3 by Obruca et al. [16] for progestin treatments. In fact, the high dose progestin results in total contraception, which means that hormonal levels are at steady state because the solutions to the differential equations are constants. Also, low and high doses of exogenous estrogen reduce the LH surge to non-ovulatory levels (Figs 6 and 7).

   

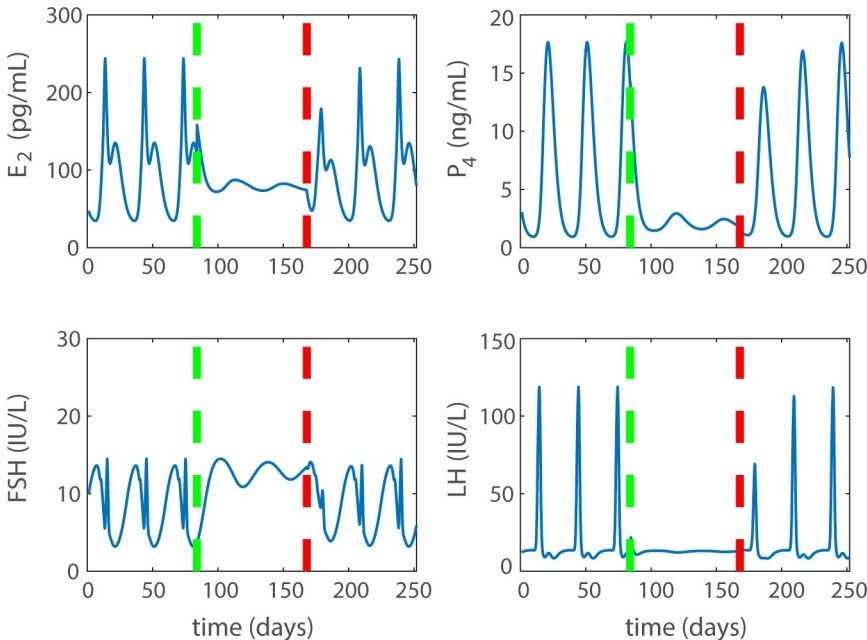

**Fig 10. Temporary dose.** Simulation (solid blue line) of a temporary treatment of a low dose combined hormonal contraceptive. Dosing begin at day 84 (green dashed line) and ends at day 168 (red dashed line) at which point the dose decreases exponentially due to the half-life of the drug. A nearly instant contraceptive effect after dosing is observed and, once the drug is removed, return to ovulation occurs within 1-2 cycles.

The high dose estrogen results in steady state hormone levels. When low dose estrogen and low dose progestin are administered together, this combined hormonal treatment achieves total contraception (Fig 8) and is compared with clinical data from [18].

In order to determine which dosing pairs result in total contraception, a Hopf bifurcation curve (Fig 9) is drawn in the ($e_{dose}$, $p_{dose}$) plane which separates the plane into a region of steady state solutions and a region of non-constant periodic solutions. For dosing pairs near this curve, LH and P4 levels are low so the menstrual cycles are non-ovulatory. As both dosing amounts decrease, the LH surge increases and the contraceptive effect is gradually lost. If a non-ovulatory LH level is assumed, the model may be used to predict which dosing pairs result in LH at or below that level and, hence, are contraceptive. In clinical settings, the contraceptive effect is reached well before the "total contraception" described by the model. Therefore, the modeling results cannot directly be translated to clinical applications, but if combined with a PKPD model, the ideas put forward here have potential to be used in designing new treatment strategies.

Also, model simulations indicate how quickly a combined contraceptive treatment produces a non-ovulatory menstrual cycle and how fast the cycle returns to normal after the treatment ends. For example, Fig 10 shows that the treatment pair ($e_{dose}$, $p_{dose}$) = (40 pg/mL, 0.6 ng/mL) results in a contraceptive state in the first cycle after dose application and an ovulatory cycle returns within one or two cycles after the treatment ends.

A limitation of the current model is the assumption that the effect of estrogen on progesterone can be combined into one term $P_{app}$ that does not differentiate the neuroendocrine verses the ovarian systems, and that $P_{app}$ has a maximum of P4. In reality the effect likely differs between organs, and it may be that for large concentrations of E2 the $P_{app}$ is larger than P4, but without additional data we chose this unifying approach. Future studies should explore these possibilities in more detail.

Another limitation is that the predicted FSH response to hormonal treatment. Eq (3) indicates that FSH synthesis only depends on inhibin A. When a contraceptive state is reached, luteinization does not occur and so inhibin A is diminished. Because inhibin A inhibits FSH synthesis, the model predicts that in a contraceptive state FSH is produced at a high level. However, this is not observed biologically [18]. To improve the model, Eq (3) needs to be modified so that FSH synthesis depends on more reproductive hormones especially inhibin B.

Finally, for each synthetic exogenous hormone, the addition of a model accounting for its pharmacokinetics and its specific activity in relation to the corresponding natural hormone would allow for more detailed representation of different treatments. While the addition of a model of this type can give insight into specific treatments, it does not change the main conclusions that contraception can be achieved at a lower dose for the combined treatments.

In summary, this study presents a mathematical model which accurately predicts daily hormone levels (LH, FSH, E2, P4, InhA) for normally cyling women. By adding two ovarian autocrine effects of E2 and P4 and only four new parameters to previous models [12–15] we have studied, this new model illustrates that progestin and synthetic estrogen treatments result in contraception. When coupled with a PKPD model for oral contraceptive drugs, the resulting model may help discover minimal effective doses of these drugs and may lead to patient-specific dosing strategies.

## Acknowledgments

For this study, we would like to acknowledge Drs. Tjeerd Korver and Michelle Fox for their scientific input and advice while the model was being developed and refined. We would also like to thank the two referees whose suggestions have significantly improved this manuscript. This work was developed in collaboration with Merck & Co., Inc., Kenilworth, NJ, USA, where Andrew Wright held an internship.

## Author Contributions

**Conceptualization:** A. Armean Wright, Ghassan N. Fayad, James F. Selgrade, Mette S. Olufsen.

**Data curation:** A. Armean Wright, Ghassan N. Fayad, Mette S. Olufsen.

**Formal analysis:** A. Armean Wright, Ghassan N. Fayad, Mette S. Olufsen.

**Investigation:** A. Armean Wright, Ghassan N. Fayad, James F. Selgrade, Mette S. Olufsen.

**Methodology:** A. Armean Wright, Ghassan N. Fayad, James F. Selgrade, Mette S. Olufsen.

**Project administration:** Ghassan N. Fayad, James F. Selgrade, Mette S. Olufsen.

**Software:** A. Armean Wright.

**Supervision:** Ghassan N. Fayad, James F. Selgrade, Mette S. Olufsen.

**Validation:** A. Armean Wright, James F. Selgrade, Mette S. Olufsen.

**Writing – original draft:** A. Armean Wright, Ghassan N. Fayad, James F. Selgrade, Mette S. Olufsen.

**Writing – review & editing:** A. Armean Wright, Ghassan N. Fayad, Mette S. Olufsen.

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
