## [Decision Letter · Decision Letter 0]

3 Oct 2019

Dear Dr Olufsen,

Thank you very much for submitting your manuscript 'Mechanistic model of hormonal contraception' for review by PLOS Computational Biology. Your manuscript has been fully evaluated by the PLOS Computational Biology editorial team and in this case also by independent peer reviewers. The reviewers appreciated the attention to an important problem, but raised some substantial concerns about the manuscript as it currently stands. While your manuscript cannot be accepted in its present form, we are willing to consider a revised version in which the issues raised by the reviewers have been adequately addressed. We cannot, of course, promise publication at that time.

Sincerely,

Eric A Sobie

Guest Editor

PLOS Computational Biology

Mark Alber

Deputy Editor

PLOS Computational Biology

[LINK]

As you will see from the reviews, both reviewers commented positively on the potential utility of the model you have developed and the importance of the questions this model can address. However, both reviewers felt that the some of the assumptions made in the development of the model could be better explained in the manuscript. Reviewer 1 also raised an important point about the quantitative comparison with experimental data. Revising the manuscript with these reviewer comments in mind is likely to lead to a stronger manuscript.

Reviewer's Responses to Questions

**Comments to the Authors:**

Reviewer #1: General Impressions

Although the results of the present work appear to qualitatively mimic those of experimental work, I believe that a major shortcoming of the submission is the absence of pharmacokinetic/pharmacodynamic considerations outside of the last paragraph of the manuscript. Given regular dosing (daily or weekly), we would expect the transient time of to contraception to be longer, given the time it would take a standard PK model to approach (an oscillatory) steady state. At the very least, the authors should put more thought into discussing the implications of this based on either preliminary results or by running additional simulations to support their claims. The treatment protocol meant to be reproduced consisted of 21 days of daily treatment, whereas the model requires constant dosing for 3 months prior to the start of the simulations, presumably to give the model time to approach a stable limit cycle or steady state. It is difficult to consider the results of the current work as a quantitatively valid approach to comparison with the treatment data, when the assumed treatment protocol is very different from that of the experiment. I recommend the authors seriously consider the discrepancy and attempt to incorporate a the more realistic treatment prior to comparing to the data. Without this final comparison, the work appears to be purely theoretical in nature. If this is the goal of the manuscript, then this should be very plainly stated.

The question of variability in contraceptive dosing is an interesting one, especially where hybrid treatments are concerned. On the one hand, the approach used by the authors--modifying an existing framework to account for additional roles for relevant hormones--is appropriate to study such a problem. However, it seems the description of the approach and explanation of why certain modeling choices were made are lacking in this manuscript. Descriptions of the work, beginning with the "Modeling contraception" section should be written in a more detailed and clear way, especially as far as modeling assumptions and their justification, and methods in exploring the edose-pdose parameter space are concerned. The organizational structure of the paper works as is, but would benefit from a great deal more detail about the simulation methods--in addition to the assumption previously mentioned--particularly with respect to distinguishing between previously used and new parameters, parameter estimation (if applicable), and the major differences between the approach in the paper compared to that of the experiment.

In addition, the manuscript would benefit from additional editing for typos, grammar, and general flow of the writing. There are also several terms that are used with inconsistent punctuation (hyphenated, etc.) and improper use of open quotation marks, which I assume are due LaTeX's idiosyncrasies.

Specific Major Comments

- Page 12-13

- Eqn (5): It is not clear why the choice is made to separate b and c1 as parameters, particularly when they are both multiplied by the FSH variable and scaled by a function of Papp. Why not just define a single parameter equivalent to b+c1, since these would not be independently identifiable anyway?

- Eqn (16): There appear to be two primary effects of exogenous progestin on the system: (1) the neuroendocrine effects involved in regulating LH synthesis and (2) the ovarian autocrine effect of reducing FSH sensitivity of follicles. The authors do not provide sufficient justification for making the modeling choices reflected in Equation (16). In particular, if E2 is meant to upregulate P4 receptors, effectively increasing P4 function, there is no differentiation between neuroendocrine and ovarian mechanisms. That is, the choice of E2-mediated feedback in Eqn 16 seems to reflect how E2 behaves in the hypothalamic-pituitary axis rather than in ovarian tissues (looking at the forms used in the original model). If the feedback modeled is universal in all tissues, then this should be stated explicitly in the manuscript.

In addition, if E2 primes P4 receptors for the luteal stage, why would this effect be reflected in follicle growth in the same way that P4 works to inhibit gonadotropins when elevated throughout the luteal stage? Is there a good explanation as to why E2-dependent Papp might affect different tissues from regularly circulating P4? I suggest that the authors explore this possibility for the sake of completeness.

It is also not clear why the priming effect of E2 can only maximally double the effective P4 concentration. Since explicit receptor expression and binding kinetics are not modeled here, it is unclear how the model chosen is mechanistically appropriate. The most general model for Eqn (16) might have the form Papp = (P4/(1+Z))*(1+ Z*Hill(E_2)), where Z is not necessarily 1 (as is currently the case). Have the authors already explored this possibility? If so, this should be mentioned in the manuscript, with a justification of the final form for Equation (16). If not, it may be useful to explore the dynamics of the model for various scalings.

Specific Minor Comments

1 - Page 3, Table 1: Are there references for the parameters used, especially those that are part of the modification? Or if most parameters used come from existing models, please also mention this.

2 - Page 5, line 99: The wording for "not well understood mechanisms" is awkward. Please consider fixing this.

3 - Page 7, lines 134-135: Are the functions of LH mentioned here also normal functions of endogenous progesterone?

4 - Page 9, Fig. 2: The hormones listed in the caption and appearing in the blue boxes include P2. Is this supposed to be P4 instead?

5 - Page 10, Eqns (3) and (4): There appears to be a typo in the denominator of the FSH release terms (1 + c_FSH,E * E_2), where E_2 should be squared. The units for c_FSH,E appearing in Table 1 are consistent with the Schlosser/Selgrade model, but this term is not.

6 - Page 11, lines 217 and 218: The description of estrogen and progestin doses being added lacks subject-verb agreement. Please check the specifics.

7 - Page 13, line 243: The FSH-dependent growth term mentioned is not actually "scaled by a factor proportional to applied progestin" as it appears in Eq (5). The terminology should be fixed in this sentence to match what is reflected in the equation in question.

8 - Page 13, lines 263-264: Statement beginning with "Insufficient gonadotropins..." is incomplete (fragment).

9 - Page 14, lines 275-276: Were any parameters estimated in the model fitting process? If so these should be explicitly indicated, either in the table of parameters or in the Results section. If not, the authors should give a description of how new parameters for the model addition were chosen.

10 - Page 15, Fig. 3 & description: How many cycles of data were used in the model fitting? What is the cycle length from the new model without contraceptives compared to the data? Is there any insight to the discrepancy between the model and the data? How does the baseline model (no estrogen/progestin) compare to the original model without the new modifications? An illustration of this might be useful for the average reader.

11 - Page 21, Fig. 9: "Low Dose Combination" is spelled incorrectly. Please add units to the axes in the graph as well. Also, it is not at all described how the bifurcation diagram was generated. Was this through ad hoc methods or using some other software? Please clarify. Further, because the manuscript places emphasis on the distinction between total contraception and "partial" contraception with periodic solutions, it would be useful to overlay regions within the periodic solutions for which some level of contraception is obtained (clinical criterion), even if this is not valid mathematically. This also seems relevant in the context of limiting contraception doses to clinically effective values, rather than mathematically effective values. More emphasis on the physiological and clinical relevance would be helpful in the discussion of model results.

12 - Page 22, lines 343-344: Please provide references for the half-lives of progestin and estrogen. Also, are the half-lives of exogenously administered hormones different from endogenous ones?

13 - Page 24, line 388: The authors comment on the shortcoming of the model, which may require that FSH synthesis be modeled using additional reproductive hormones. I assume the authors are referring to Inhibin B, among others. If this is the case, the known hormones should be stated explicitly.

14 - General: In the Results section, it is left unclear whether the doses used in the simulations are used to match the data in reference [15] or whether they reflect the minimum doses required to elicit total (or partial) contraception. The bifurcation diagram certainly matches these values, but it is unclear whether this is a by-product of the model itself or whether the parameters were tuned to match experimental data. Please clarify.

Reviewer #2: Review of PCOMPBIOL-D-19-01189, Mechanistic model of hormonal contraception

General comments

The paper describes an augmentation of a previously published model of hormonal regulation of the menstrual cycle, through addition of two autocrine terms, and demonstrates model predictions given low and high dosing of estrogen alone, progesterone alone (nominally as progestin), and a combination of the two. The model has potential utility in designing hormone dosing regimens to achieve birth control, characterized by an acyclic state with low LH throughout, and also to elucidate what mechanism of the female reproductive cycle are key to explaining observed responses to birth control regimens. Unfortunately, the paper as is lacks sufficient detail to achieve either of these, but I believe that a small amount of additional work could help it to meet these objectives. I also suggest ways in which the results could be significantly improved, but those could be discussed as potential future research.

For context, I have experience in the mathematical modeling of endocrine/hormonal systems, but not in contraception science. One way in which the model could be useful is that researchers seeking to refine birth control methods (combinations of estrogen and progesterone capable of achieving birth control) might use it to identify regimens that will achieve this purpose. But to be convinced that it can be used for this purpose, I would want to know that it can accurately predict the result of existing regimens. The authors demonstrate an ability to simulate partial and complete birth control, but how do the doses used in the modeling that achieve these effects (low and high estrogen, for example) compare to corresponding clinical doses? There is a brief reference to progestin levels in [15], made towards the end of the introduction, but the specific levels used in those experiments and the current simulations (and rationale for them) should be described in the methods section on modeling contraception. Further, the P4 and FSH predictions in Figure 5, for example, are well outside the observed ranges. Off hand these decrease my confidence in the model’s ability to predict hormonal response to contraceptive dosing. At a minimum, I suggest that the authors discuss how the simulated estrogen and progesterone (progestin) compare to doses used in the study which they simulate. Better would be to explore what model parameters or features would need to be adjusted such that all predicted hormone levels are within the experimental range from [15], Obruca et al. (2006). A rough sensitivity analysis could provide the later.

A very interesting qualitative result stated in the paper, but not really shown, is that two autocrine terms had to be added to the ovarian model in order for the birth control regimens to be successfully described. But the authors don’t show predictions of the original model, without these terms: how badly does the original model fail to predict contraceptive dosing response? What happens if only one of the autocrine terms is added? How did the authors determine that both were necessary? In short, show me that *both* additional terms are needed for the model to perform successfully, by showing how it fails without them, or with only one of them.

Also, please provide references to biological sources which describe these autocrine mechanisms… reference for the effect of progestins on follicle sensitivity to FSH [15, 19] are given in the “Contraceptive mechanisms” section, but these should be repeated in the “Ovarian model” section, where the mode equations are being described. I did not see references for the autocrine effect of estrogen. The supporting literature/data should be cited along with the description of the corresponding equations.

Also, how where the forms and the parameters of the autocrine terms selected? They are fairly standard Hill-like forms, but how were the values of the coefficients obtained? It’s fine if this was a simple attempt to fit data, but then that should be stated. Where they arrived at by trial-and-error, or formal optimization? Implicit in the estrogen autocrine term (equation 16) is that the effect varies between a maximum of 1 (100%) and 0.5 (50%). How was this range identified? Are there supporting data? Were other ranges from minimal to maximal induction considered?

Specific comments

While it is nice to augment figures with color, keep in mind that 10% of people (readers) are color blind. Different line types (short dash, long dash, dot-dash, etc.) and/or colors that are distinguishable when printed in grey-scale should be used.

P. 5, first line: *usually* a single follicle will advance to ovulation. Sometimes it’s more than one! (This is said later but should be said here.)

The term “applied progesterone” (for equation 16, first introduced at the bottom of p. 11) lacks biological meaning. The specific autocrine effect being described is induction of the progesterone receptor. A separate term, really equation (16) but for P4, should be defined as the scaled progesterone receptor (PR) expression. Since the maximum of this term is 1, it PR expression / maximal PR expression. The degree of PR activation is then defined as the product of P4 and PR expression. This product could also be called progesterone signal.

I checked the cited references for the original model [12, 13] and another by those authors (Clark et al., 2003) and saw that that version of the model better replicated FSH peak, which this version appears to badly under-predict (Figure 3), and the height of the secondary E2 surge. I suspect that when the autocrine terms were added, the original model parameters were not re-tuned to match these features of the normal cycle. When the model structure was changed, the model parameters should have been revised. Is the low FSH peak (relative to the original model) due to the autocrine terms? Also, how is it that the autocrine terms do not extinguish the normal cycle entirely, without exogenous hormones?

And it would be much better if the experimental data in Fig 3 were shown with confidence bounds (error bars), since that would allow a reader to judge whether areas where the fit is poor are at least within the range of variability.

Fig 6: the legend says that the LH surge has been eliminated, but there is still a small peak occur ~ every 33 days. I would say “significantly suppressed” instead.

Fig 9: again, I’d like to know how the P4 and E2 doses used at the three dose levels compare to those actually used.

**Have all data underlying the figures and results presented in the manuscript been provided?**

Reviewer #1: Yes

Reviewer #2: No: When I click on the 'view submission' link, I just get the pdf of the paper, figures, tables. If spreadsheets with the data and computational results are available, I would have to go searching for them.

PLOS authors have the option to publish the peer review history of their article (what does this mean?). If published, this will include your full peer review and any attached files.

Reviewer #1: No

Reviewer #2: No

---

## [Decision Letter · Decision Letter 1]

3 Apr 2020

Dear Prof. Olufsen,

We are pleased to inform you that your manuscript 'Mechanistic model of hormonal contraception' has been provisionally accepted for publication in PLOS Computational Biology.

Best regards,

Eric A Sobie

Guest Editor

PLOS Computational Biology

Mark Alber

Deputy Editor

PLOS Computational Biology

The two reviewers felt that your revisions improved the manuscript substantially. As you will see, Reviewer 2 made suggestions of a few additional minor changes that you can consider before uploading final documents.

Reviewer's Responses to Questions

**Comments to the Authors:**

Reviewer #1: The author has addressed all previous comments satisfactorily. No additional comments.

Reviewer #2: The paper is much improved and could be accepted as written, however I have a few suggestions to be considered.

First, the authors state that to include the PK of the exogenous hormones used for birth control, individual ("patient-specific) data would be needed. I disagree. The rest of the model is built to describe an average or typical patient, and the parameters of the ovarian and HP models are not patient-specific, so there's no reason that population-average PK parameters couldn't be used for estrogen and progestin to make that part of the model more realistic.

While the functional form of estrogen's effect on P4 sensitivity is realistic, represents qualitatively what might result from a model of progesterone receptor activity, it is quite empirical, especially assuming the same response in the follicles as in the hypothalamus/pituitary. Also, I would assume that the basal level in the absence of E2 is "1" and that it increases from there, but the model as is is tuned fairly well to the normal cycle, so there would only be a nominal impact of such a change, as other parameters would need to be changed to offset this change in the P_app term.

Fig 2 legend, 3rd sentence, "The black horizontal arrows represent hormone movement...." This is not true for the ovarian model. There the black arrows represent movement of cells or tissue between stages, which have differing hormone production capacity.

P. 11, line 207, "The Hill function is the main biological mechanism..." I would restate this. A function is just a mathematical construct, it can't be a biological mechanism. But it can represent a mechanism. Can you briefly say what biological mechanism is being represented and why the term is so important to the model?

**Have all data underlying the figures and results presented in the manuscript been provided?**

Reviewer #1: None

Reviewer #2: Yes

PLOS authors have the option to publish the peer review history of their article (what does this mean?). If published, this will include your full peer review and any attached files.

Reviewer #1: No

Reviewer #2: No

---

## [Editor Report · Acceptance letter]

11 Jun 2020

PCOMPBIOL-D-19-01189R1 

Mechanistic model of hormonal contraception

Dear Dr Olufsen,

I am pleased to inform you that your manuscript has been formally accepted for publication in PLOS Computational Biology. Your manuscript is now with our production department and you will be notified of the publication date in due course.

With kind regards,

Sarah Hammond
